# The Route of Administration of Rabies Vaccines: Comparing the Data

**DOI:** 10.3390/v13071252

**Published:** 2021-06-27

**Authors:** Deborah J. Briggs, Susan M. Moore

**Affiliations:** Department of Diagnostic Medicine/Pathobiology, College of Veterinary Medicine, Kansas State University, Manhattan, KS 66506, USA; smoore@vet.ksu.edu

**Keywords:** rabies, post-exposure prophylaxis, PEP, pre-exposure vaccination, PreP, immunogenicity, efficacy, anamnestic response

## Abstract

Cell culture rabies vaccines were initially licensed in the 1980s and are essential in the prevention of human rabies. The first post-exposure prophylaxis (PEP) vaccination regimen recommended by the World Health Organization (WHO) was administered intramuscularly over a lengthy three-month period. In efforts to reduce the cost of PEP without impinging on safety, additional research on two strategies was encouraged by the WHO including the development of less expensive production methods for CCVs and the administration of reduced volumes of CCVs via the intradermal (ID) route. Numerous clinical trials have provided sufficient data to support a reduction in the number of doses, a shorter timeline required for PEP, and the approval of the intradermal route of administration for PEP and pre-exposure prophylaxis (PreP). However, the plethora of data that have been published since the development of CCVs can be overwhelming for public health officials wishing to review and make a decision as to the most appropriate PEP and PreP regimen for their region. In this review, we examine three critical benchmarks that can serve as guidance for health officials when reviewing data to implement new PEP and PreP regimens for their region including: evidence of immunogenicity after vaccination; proof of efficacy against development of disease; and confirmation that the regimen being considered elicits a rapid anamnestic response after booster vaccination.

## 1. Introduction

Rabies is a zoonotic disease with up to 99% of human rabies deaths caused by exposure to infected dogs [1]. Consequently, the most efficient global strategy to prevent most human rabies cases is to eliminate the spread of rabies at the source of infection, i.e., in dogs. Several successful mass canine vaccination programs in canine rabies endemic countries have proven that elimination of canine rabies is feasible, and it is cost effective in preventing human rabies deaths [2,3,4]. However, until global elimination of canine rabies has been achieved, improving access to human rabies vaccines in canine rabies endemic countries is key for reducing the current number of human rabies deaths. In fact, after an exposure to a rabid animal has occurred, administration of human rabies vaccines, along with proper wound care, is considered to be one of the first lines of defense in preventing human rabies.

Cell culture rabies vaccines (CCVs) are among a select few human vaccines that can protect against disease after an exposure to a pathogen has occurred [5]. Post-exposure prophylaxis (PEP), consisting of the administration of multiple doses of vaccine administered over a specific time period, has been so successful in protecting humans against rabies after an exposure has occurred that it is not typically necessary to prevaccinate an entire population in rabies endemic regions in order to prevent rabies.

CCVs, developed over four decades ago, are among the most effective human vaccines ever developed and have transformed the way that public health officials create and implement human rabies prevention programs [6,7,8]. The most widely available CCVs used today are produced in one of three types of cells including: human diploid cells, vero cells; and chick embryo cells. At the time of this publication, there are four CCVs that have met the WHO prequalification standards [9].

The recommendations for administration of human rabies vaccines necessarily vary from country to country. For example, one country may have determined that it is beneficial to use the ID route of administration, whereas another country does not. A review of the published literature reveals that there have been numerous clinical trials conducted to evaluate the IM and ID route of administration in various regimens [10,11,12,13]. However, the sheer abundance of the published data associated with the IM and ID routes of administration can make it difficult to compare these data when attempting to make changes to national rabies vaccination regimens for human rabies prevention. This brief review examines three critical factors that should be included in the evaluation process when it is necessary to compare clinical data from IM and ID administration of CCVs. These factors include evidence of rabies virus neutralizing antibodies (RVNAs) after vaccination, proof that the regimen under consideration protects patients exposed to confirmed rabid animals, and confirmation that an anamnestic response to booster vaccination occurs in persons that have received pre-exposure prophylaxis (PreP).

## 2. Comparing the Data

Clearly, national health agencies wishing to institute new or update their current recommendations for human rabies prevention need to consider and evaluate a wide range of data. However, as mentioned above, the plethora of published information using different vaccines, routes of administration, and schedules can make direct comparison of data difficult and complex. Currently, there is no precise scientific method to confirm absolute equivalence in every possible immunological aspect when comparing data from IM regimens vs. ID regimens. However, three criteria can serve as confirmatory “benchmarks” of equivalence when comparing peer reviewed published clinical data from rabies IM and ID vaccination trials. These three criteria include: (1) immunogenicity, i.e., to provide evidence of the production of RVNAs after vaccination; (2) efficacy, i.e., to prove that a new regimen protects patients that have been exposed to laboratory confirmed rabid animals; and (3) anamnestic response, i.e., to ensure that when a previously vaccinated patient receives a booster dose of vaccine, the patient’s immune system immediately begins to produce RVNAs (Table 1). For the purposes of this review, we examined these three benchmark criteria using published data from peer reviewed clinical trial reports that evaluated rabies vaccines that have met the WHO criteria for prequalification [9].

### 2.1. Immunogenicity

The production of RVNAs after PEP and PreP is a vital component of a successful immune response after vaccination. In fact, the recommendations for administering PEP to patients exposed to suspected or confirmed rabid animals includes procedures to ensure that RVNAs (in the form of rabies immunoglobulins (RIGs)) are administered as ‘passive immunity’ at the time of, or up to 7 days after, the first dose of vaccine for PEP is given [14,15,16,17]. Passive immunity, in the form of RIGs, helps to reduce the risk of disease progression by making RVNA available at the wound site as soon as possible to eliminate rabies virus that may have been injected during the trauma of exposure. RIGs are not administered more than 7 days after the first dose of vaccine was administered because after that point in time, a patient’s own immune system has initiated the production of RVNAs (active immunity). It generally takes from 7–14 days for a patient’s immune system to produce a detectable level of circulating RVNAs following the first dose of vaccine.

For reasons stated above, immunogenicity is the first step in proof of concept for a proposed new PEP regimen. The presence of detectable RVNA on or before Day 14 is considered to be proof that the PEP regimen under consideration produces a robust immune response. There is no titer value that has been proven to be ‘protective’ in humans. It is clear that to conduct a clinical trial in which humans were challenged with rabies virus to identify an exact protective antibody level would be unethical. However, a serological titer of 0.5 IU per mL on Day 14 after the initiation of vaccine is considered to be adequate proof of immunity. When testing for RVNA values, it is important to utilize an assay that actually measures neutralizing antibody rather than binding antibody. The rapid fluorescent focus inhibition test (RFFIT) and fluorescent antibody neutralization test (FAVN) are both appropriate for this purpose [18,19,20,21]. Most clinical trials evaluating the efficacy of a new or alternative regimen also include protocols to evaluate levels of RVNA at various times points throughout the clinical trial to track the kinetics of the immune response. The four CCVs that are currently listed as the WHO prequalified have all conducted clinical trials and published data proving that they do produce a robust immune response after IM PEP and ID PEP and confirming that RVNA is present by Day 14 after initiation of vaccination [11,13,22,23,24].

PreP is recommended for persons whose occupation or hobbies puts them at increased risk of exposure to rabies, and therefore is administered before an exposure occurs. Although it is important to confirm that a new PreP regimen elicits the production of RVNAs, the objective of PreP is not to produce RVNA as quickly as possible in response to a recent exposure. Rather, the objective of PreP is to eliminate the need for administering RIG and to prime the immune system so that it elicits a rapid immune response to a booster vaccination in the event that a future exposure to rabies [17,25]. Therefore, some studies examining new PreP regimens have not included a blood sample on Day 14 but have withdrawn a blood sample later in the timeline of the clinical trial, for example on Day 35. In examining new PreP regimens, proving that that the vaccine or regimen elicits an immune response is the first step in confirming that a prevaccinated person also has a rapid anamnestic response when given a booster dose or series of doses of vaccine. In PreP clinical trials, blood draws between Day 14 and 35 after initiation of vaccination and assayed to confirm the presence of RVNA can provide the evidence needed to confirm that the regimen under consideration is immunogenic. The immune response of PreP administered by the IM and ID route have been compared for several regimens in many studies and the WHO has updated their recommendations for PreP as new data have become available [26,27,28]. Currently, the WHO recommendations for PreP include a two dose one week ID PreP and a three dose ID or IM PreP regimen administered over a period of 21–28 days [1].

### 2.2. Efficacy

Proving the efficacy of a newly proposed PEP regimen is of paramount importance for obvious reasons. Conducting a clinical trial to prove efficacy must only be initiated after proof of immunogenicity is confirmed. When conducting a clinical trial to prove efficacy, it is important to be able to collect diagnostic samples from the animal (i.e., brain tissue) involved in the exposure and submit them to a reliable rabies diagnostic laboratory to confirm that the animal was rabid. The first field trial to prove efficacy of HDCV was conducted in Iran between 1975 and1976 and involved 45 patients bitten by confirmed rabid wolves and dogs [6]. Forty-four patients also received anti-rabies serum. One patient, exposed to aerosol rabies, was not given the anti-rabies serum. Six doses of HDCV were administered subcutaneously and all patients survived proving that HDCV was effective and would prevent disease in patients exposed to rabies.

Administering CCVs by the ID route also required proof of safety and effectiveness. One of the first clinical studies to investigate the efficacy of ID PEP was conducted in Thailand by Warrell et al. and included 0.1 mL doses of HDCV administered at eight sites on Day 0, four sites on Day 7, one site on each of Days 21 and 91 [29]. All patients survived exposure to confirmed rabid animals. This study and the study also included serological evidence of immunogenicity. Further clinical trials conducted in Thailand examined reduced ID PEP regimens in patients exposed to suspect and later confirmed rabid animals [30,31]. The clinical trial conducted by Chutivongse et al. enrolled 100 patients that had been severely bitten by confirmed rabid animals and all patients were followed for one year after vaccination. All patients were confirmed to be alive one year after the final dose of vaccine was administered. These initial studies provided the proof needed that ID PEP was as effective as IM PEP when administered according to the schedules administered in the clinical trials and have served as models for designing additional clinical trials evaluating the effectiveness of new PEP regimens.

The ID PEP study by Phanuphak and colleagues briefly described above was confirmed in a second study conducted in Thailand by Chutivongse et al. [30,31]. This second study enrolled 100 patients that had been severely bitten by confirmed rabid animals and all patients were followed for one year after vaccination. All patients were confirmed to be alive one year after the final dose of vaccine was administered. These initial two efficacy studies provided the proof needed that ID PEP was as effective as IM PEP when administered according to the schedules listed above and have served as models for designing additional clinical trials evaluating the effectiveness of new PEP regimens. Further clinical studies have confirmed that the other WHO prequalified rabies vaccines were also safe and effective for both IM PEP and ID PEP [10,13,24,32,33,34,35,36].

### 2.3. Anamnestic Response

It is clear that PEP and PreP are both effective strategies to protect a person at risk of contracting rabies. However, the approach to reach protection from disease after PEP and PreP differ. Proving that an anamnestic response occurs after PEP is not a primary objective, whereas it is one of the two basic reasons for a person at higher risk of exposure to rabies to receive PreP. PreP is administered in order to ensure that a vaccinated person can mount a rapid anamnestic response to booster vaccination in the event of a future exposure. There are several studies that have evaluated and confirmed evidence of an anamnestic response after booster in patients that have received PreP by IM and ID routes of administration, including up to 10 to 24 years after receiving the initial PreP series [37,38,39,40,41]. It is important to withdraw a blood sample prior to administering the booster dose or doses of vaccine in order to quantify the baseline titer prior to booster. After the booster has been administered, a second blood sample should be withdrawn on Day 7–10 to determine that a rise in titer has occurred, thus, confirming that anamnestic response has occurred.

## 3. Conclusions

The role of CCVs in preventing deaths and transforming public health strategies to prevent human rabies cannot be overstated. If the initial lengthy, and costly, vaccination regimens using CCVs for PEP had remained unchanged, it is highly likely that governments would not have had the incentive to replace NTVs in a timely manner. As discussed in this paper, numerous clinical trials conducted over the past four decades have investigated several variations of shortened vaccination regimens for both PEP and PreP. The results of those clinical trials have led to the accumulation of a considerable amount of published data and attempting to compare different IM and ID regimens can be a daunting task for governmental health agencies aiming to improve access to rabies biologicals in their country. However, the surplus of published clinical data can also be an invaluable source of information when making decisions regarding national recommendations for PEP and PreP regimens by focusing on three specific criteria to serve as benchmarks in equivalence. These include evidence of immunogenicity at a specific time after initiation of vaccination, proof of efficacy after exposure to rabies virus, and establishing the fact that the new regimen/vaccine induces a rapid anamnestic response to a booster vaccination.

## Figures and Tables

**Table 1 viruses-13-01252-t001:** Immunogenicity, efficacy, and anamnestic response. Suggested minimum number of blood draws required to confirm proof of immunogenicity after intramuscular (IM) or intradermal (ID) post-exposure prophylaxis (PEP) or pre-exposure vaccination (PreP); suggested confirmatory evidence of protection against rabies after proven exposure to rabies; and suggested proof of an anamnestic response after booster in persons that have previously received PreP by either IM or ID route. Rabies virus neutralizing antibody (RVNA) levels should be evaluated using a validated RVNA assay. NA refers to Non-Applicable.

Criteria	Required Data		Supplemental Data
	Minimum Number and Suggested Time Period of Blood Draw after Initiation of PreP or PEP to Confirm RVNA Response	Minimum Time Period after Initiation of PEP to Confirm Patient Has Survived Exposure to Confirmed Rabid Animal	Considerations for Additional Blood Draw Dates
**1.** **Immunogenicity**	PrEP (ID or IM): Day 0 and 35 PEP (ID or IM): Day 0, 14, and 28	NA	Supplementary blood draw dates will provide additional data points to monitor the rise and decrease in titers over time.
**2.** **Effectiveness**	PEP (ID or IM): Day 0, 14 and 28	PEP (ID or IM): One year after first dose of PEP administered	Including blood draws as part of a clinical trial to test efficacy is helpful to confirm immune response after PEP.
**3.** **Anamnestic Response**	One year after first dose of PreP series (ID or IM) was administered to document baseline titer prior to booster, and 7 to 14 days after administration of booster dose(s)	NA	NA

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
