# Peer review of "The Route of Administration of Rabies Vaccines: Comparing the Data"

_viruses, 2021, doi:10.3390/v13071252_

Round 1
Reviewer 1 Report
I have no major concerns with the manuscript.
Minor comments
l. 157-158 consider rewording
l. 187-188 sentence is duplicated
Author Response
Thank you for the review. I have reworded the sentence about a "protective" level of rabies neutralizing antibodies and deleted the duplicated sentence as suggested by the first reviewer.
Reviewer 2 Report
This is an interesting paper that compares the protocols of administration of cell culture rabies vaccines currently recognized by WHO. It examines three critical benchmarks that can serve as guidance for health officials when reviewing data to implement new PEP and PreP regimens for their region: evidence of immunogenicity after vaccination; proof of efficacy against development of disease; and confirmation that the regimen being considered will elicit a rapid anamnestic response after booster vaccination.
Minor revisions are included in the text.
Line 164: Concerning FAVN test, I would suggest the following reference:
Cliquet F, Aubert M, Sagne L Development of a fluorescent antibody virus neutralisation test (FAVN test) for the quantitation of rabies-neutralising antibody. J Immunol Methods, (1):79-87, 1998.

Author Response
Thank you for the comments. I have added the appropriate reference for FAVN by Clique, Aubert and Sagne as suggested.
Reviewer 3 Report
The authors state that the introduction of cell culture vaccines (CCVs) for rabies post-exposure prophylaxis (PEP) has profoundly altered the ability to prevent human rabies. They explain that schedules have changed over the years and IM regimens may be best in some countries (e.g., ones where a large number of people are not vaccinated at once), while ID schedules are essential in other countries (e.g., cost-savings). Because of the diversity of vaccine series currently in vogue and the lack of a scientific method to confirm absolute equivalence, the authors, propose criteria use of 3 criteria (i.e., immunogenicity, efficacy, and anamnestic response) as a benchmark of equivalence when comparing clinical data from IM and ID vaccination trials. They provide a roadmap for using these 3 criteria.
Overall impression: The problem posed by the authors is indeed a real problem—many vaccination schedules exist but readers may not know how to forecast appropriateness of an ID schedule based on IM trial data and vice versa. Public health bodies developing post-exposure prophylaxis recommendations may be confronted with interest in developing evidence-based guidelines but being unsure how to extrapolate data about one series to another. The organization of the paper into sections about each of these criteria effectively outlines how these can be used, for example, when clinical trials involve a blood draw at day 35 instead of day 14. The three measures discussed in this paper will be invaluable to WHO/SAGE and ACIP as guidelines are updated in the future.
General comments: The authors are very knowledgeable about rabies and provide a comprehensive introduction and explanation of the three criteria. There is a wealth of historical information included in the manuscript including the reason that intradermal schedules were introduced and the various schedules that have been introduced over the years. However, the purpose of the paper seems to be to provide criteria that can be used to compare IM and ID schedules for vaccine recommendations. The historical information is critical to understanding the reasons specific changes were made over the years. But in a paper about how to compare seemingly discordant schedules, the detail sometimes muddled the main message of the paper. The authors may consider publishing the historical information separately. Wherever possible, condensing the text (eg., providing a more succinct introduction by condensing the content of the last 3-4 paragraphs to simply state that there are very diverse schedules and eliminating the information about the timeline along which these were introduced, might make the main points of the articles clearer. Additionally, a table simplifying the important elements for each criteria (e.g., in a row about immunogenicity, lab tests that should be used to assess immunogenicity and the time points at which immunogenicity should be assessed are important.) Referring to the simplified information in the table might make the take-home messages easier to grasp for persons with less rabies expertise.
Specific comments:
-136 are suggested as the criteria to compare varying and IM schedules.
-Line 36: I'm not sure if vaccines are the FIRST line of defense in preventing human rabies. Avoidance of risky exposures, wearing proper personal protective equipment when working with animals etc. are probably the first line of defense. Often those are not enough and with no treatment options available, PEP is the only preventive strategy once an exposure occurs. I suggest clarifying that sentence.
-Line 44: Consider adding "pre" in the following sentence: "...it is not typically necessary to pre-vaccinate an entire population in order to protect..." You could also simply say, "vaccinate an entire population after an exposure" so that it's clear that you are referring to pre-exposure prophylaxis. Without this modification, the reads as if post-exposure prophylaxis does not involve vaccinations. Same comment for line 47.
-Line 67, the word "to" is missing from the sentence. It should say, "...in clinical trials in concerted efforts [to] the cost of PEP."
-It's unclear how #3, Vaccination failures, fits into the manuscript.
-The abstract mentions the historical context for ~2/3 of the abstract length before it mentions the objective. The authors may consider explaining the 3 proposed criteria in lieu of some of the background information.
Author Response
Thank you for the careful review. In response to your comments, we have significantly reduced the information on the history of vaccination regimens in the introduction and have corrected the spelling and grammar that was identified by the reviewer. We have also added a table, as suggested to help clarify the topic.
We believe that the paper reads better and is clearer with the suggested editorial changes.